# Effective and safe: Long-term aerosol disinfection of slightly acidic electrolyzed water causes no harm in rats

Takanori Oikawa[1], Kana Akinaga[2], Megumi Yanagida[1], Wataru Yamazaki[1], Tomoko Sato[3], Yuki Mukoda[3], Koya Terada[3], Yuriko Fujii[1], Misako Higashiya[1], Shinji Yamasaki[4,5,6], Takahiro Adachi[2], Shinsuke Seki [1]*

1 Experimental Animal Division, Bioscience Education and Research Support Center, Akita University, Akita, Japan, 2 Department of Systems Design Engineering, Akita University, Akita, Japan, 3 Research Laboratory, Local Power Co., Ltd., Akita, Japan, 4 Graduate School of Veterinary Science, Osaka Metropolitan University, Osaka, Japan, 5 Asian Health Science Research Institute, Osaka P Metropolitan University, Osaka, Japan, 6 Osaka International Research Center for Infectious Diseases, Osaka Metropolitan University, Osaka, Japan

* sseki@gipc.akita-u.ac.jp

## Abstract

Slightly acidic electrolyzed water (SAEW) has been shown to possess strong bactericidal and virucidal properties, making it a promising candidate for spatial disinfection. In this study, we rigorously evaluated the efficacy and safety of SAEW for aerosol disinfection under controlled conditions. Laser particle size distribution analysis confirmed uniform aerosolization. Additionally, analysis of chlorine concentration ensured stable disinfection conditions. Moreover, SAEW exhibited potent sterilization effects against the model organism *Escherichia coli* in both direct-contact and aerosol-disinfection experiments. Notably, long-term exposure assessments in rats revealed no adverse effects on body weight, food and water intake, and organ function and histology. Conclusively, these results indicate that SAEW is a highly effective and safe disinfectant for controlling airborne and droplet-mediated infections. In addition to preventing the spread of infectious diseases, including coronaviruses, SAEW is expected to be effectively utilized in the veterinary, agricultural, and food industries.

## Introduction

The coronavirus disease (COVID-19), first identified in 2019 [1,2], triggered a global pandemic. As of January 2025, reports from the World Health Organization (WHO) indicate that more than 700 million cumulative cases and approximately 7 million fatalities have been recorded [3]. COVID-19 continues to circulate globally, with novel variants emerging at regular intervals [4]. Elderly individuals, people with underlying medical conditions, and immunocompromised populations remain particularly vulnerable to severe disease. Persistent symptoms following infection, collectively termed long COVID, including fatigue, respiratory distress, and cognitive impairment, have

**Data availability statement:** Yes - all data are fully available without restriction; All relevant data are within the paper and its Supporting Information files.

**Funding:** "This work was supported by Local Power Co., Ltd., and by a grant from the Akita Prefectural Government under the FY2020 "COVID-19 New Normal / New Business Feasibility Study (FS) Program". The funders had no role in study design, data collection and analysis, decision to publish, or preparation of the manuscript.".

**Competing interests:** This study was conducted as a collaborative research project with Local Power Co., Ltd., which provided financial support and supplied slightly acidic electrolyzed water (SAEW, iPOSH). Some authors (T.S., Y.M., and K.T.) are employees of Local Power Co., Ltd.; however, their involvement was limited to pre-shipment performance testing of SAEW, including available chlorine concentration, pH, and basic quality assessment. All animal experiments, data collection, and evaluation of SAEW effects were conducted independently at Akita University. All analyses were performed objectively and scientifically, and this does not alter our adherence to PLOS ONE policies on data and material sharing.

also been documented [5]. The continuous risk of emerging infectious diseases highlights the need for effective disinfection strategies.

Respiratory viruses, such as coronaviruses and influenza viruses, are transmitted through contact, droplets, and airborne particles in enclosed or poorly ventilated settings. SARS-CoV-2 spreads primarily through respiratory droplets and contaminated surfaces [6]. To mitigate such infections, aerosol disinfection with a safe virucidal solution is a promising approach for neutralizing airborne and droplet-associated pathogens. Conventional disinfectants, including alcohol, chlorine dioxide, and sodium hypochlorite, are widely used for hygiene; however, their cytotoxicity renders them unsuitable for aerosol disinfection, underscoring the need for safe and practical alternatives.

Slightly acidic electrolyzed water (SAEW) is produced by electrolysis of 2–6% hydrochloric acid or sodium chloride in a non-membrane cell, with a pH of 5–6.5. This process results in strong bactericidal activity, low environmental impact, and high storage stability [7–9]. The high oxidation–reduction potential (ORP) damages microbial membranes, allowing hypochlorous acid (HOCl) to penetrate cells and exert oxidative effects, ultimately inactivating microorganisms [9]. Previous studies have shown that SAEW effectively inactivates a broad spectrum of pathogens with higher efficiency than commercial sodium hypochlorite solutions [10–12], and we previously confirmed its strong virucidal activity against coronaviruses [13]. Upon contact with organic matter, HOCl is converted into hypochlorite ions (OCl$^-$) and water ($H_2O$), reducing its bactericidal activity but generating byproducts considered non-toxic to humans, animals, plants, and food [9,14]. Compared with strongly acidic electrolyzed water (StAEW), SAEW remains effective at lower chlorine concentrations (10–30 mg/L) and near-neutral pH, minimizing residual chlorine and chlorine gas formation while ensuring long-term stability. These properties make SAEW a promising candidate for aerosol disinfection.

Despite its potential, safety assessments of disinfectants administered via aerosol disinfection in animal models remain limited, largely due to the absence of agents deemed sufficiently safe and the technical challenges inherent to aerosolized evaluation. In this study, we employed a laser particle size distribution analyzer to ensure uniform aerosolization of SAEW and used a chlorine concentration detector to verify stable conditions. To assess the sterilization efficacy of SAEW following aerosol disinfection, we selected *Escherichia coli*—a bacterium generally more resistant to inactivation than SARS-CoV-2—as the test microorganism. Because aerosol disinfection may exert systemic or cumulative effects not detected through *in vitro* assays, we performed an *in vivo* safety evaluation using rats, assessing physiological and histological parameters. Our study provides essential insights into the feasibility of using SAEW as a practical and safe method for spatial disinfection.

## Materials and methods

### Animals

Male and female Charles River outbred Sprague–Dawley rats (8 weeks old) were housed in an environmentally controlled room (temperature, 24 ± 3 °C; humidity,

$50 \pm 10\%$) under a 12 h light (07:00–19:00)/12 h dark cycle. All animal experiments were approved by the Animal Experimentation Committee of Akita University (ID: a-1–0287) and conducted in accordance with the Animal Research: Reporting of In Vivo Experiments (ARRIVE) guidelines and the university's institutional regulations. Ethical approval for this study was obtained from the Akita University Ethics Committee. All efforts were made to minimize animal suffering throughout the experiment. Animal handling and monitoring, including evaluation of activity, posture, and any abnormal signs, were performed under the supervision of Y.F., a licensed veterinarian, and T.O., a certified laboratory animal technologist instructor accredited by the Japanese Association for Laboratory Animal Science (JALAS).

## Aerosol disinfection and confirmation of SAEW spatial distribution using laser measurement

SAEW (iPOSH: Local Power Co., Ltd., Akita, Japan) used in this study is a mixture of HClO and hypochlorite ion. The HClO in iPOSH was purified via the ion-exchange method to remove residual sodium ions from the NaClO solution. Considering that some of the HClO can be dissociated to hypochlorite ion in water, iPOSH contains not only HClO but also hypochlorite ion. The pH of iPOSH was confirmed to be slightly acidic each time before aerosol disinfection (pH, $6.31 \pm 0.15$, N = 65).

Rats were temporarily housed in a custom-made, box-shaped chamber (height [H] $620 \times$ width [W] $1920 \times$ depth [D] 930 mm). Thereafter, SAEW was sprayed into the box using a mist generator (NB-2N, SHIBATA SCIENTIFIC TECHNOLOGY Ltd., Tokyo, Japan). Notably, the area where the rats were placed, and the mist was sprayed, measured H: $500 \times$ W: $1800 \times$ D: 500 mm. During the spraying period, the mist generator produced fine droplets with an average particle size of 2–5 µm. HClO solutions at three concentrations (50, 150, and 250 mg/L) were sprayed at a rate of 15 mL/h for approximately 1 h.

To monitor the airborne concentration of HClO, a chlorine gas detector (XPS-7, New Cosmos Electric Co., Ltd., Osaka, Japan), based on semiconductor sensor technology, was placed inside the chamber. Gas concentrations were measured every 10 s, and the maximum values during the exposure period were recorded.

Particle size distribution was measured after stabilization of mist diffusion using a laser diffraction particle size analyzer (FLD-319A, Seika Digital Image Co., Ltd., Tokyo, Japan) based on the Fraunhofer diffraction method. Notably, the analyzer has a wavelength of 532 nm, a measurement range of 1–500 µm, and an accuracy within 3%. Additionally, its emitter and receiver units were positioned on either side of the chamber to allow non-intrusive *in situ* measurement. An overview of the experimental setup, including the chamber, mist generator, chlorine gas detector, and particle size analyzer, is shown in Fig 1A.

## Confirmation of the sterilization effect of HClO solution

To confirm the sterilization effect of SAEW, *E. coli* was used for all experiments. *E. coli* was purchased from NITE Biological Resource Center (NBRC, #3301) as L-dried specimens. L-dried *E. coli* was revived using a revival medium for L-dried specimens (DAIGO; FUJIFILM Wako Pure Chemical Corporation, Osaka, Japan) according to the manufacturer's protocol. Revived *E. coli* was suspended in lysogeny broth (LB) medium containing 8% DMSO and stored at ‑80 °C until experiments. All bactericidal assays were performed using *E. coli* prepared from the same master frozen stock; however, each experiment used an independently prepared aliquot that was thawed and consumed in a single use. In subsequent experiments, approximately $1.0 \times 10^6$ colony-forming units (CFU) of *E. coli* was sterilized with SAEW unless otherwise mentioned.

## Assessment of the sterilization efficacy of SAEW via the direct-contact method

Briefly, the sterilization efficacy of SAEW against *E. coli* was examined using the previously described direct-contact method [13] with minor modifications. Specifically, varying concentrations (31.3, 62.5, 125, and 250 mg/L) of SAEW (693 µL) were reacted with *E. coli* suspension ($1.0 \times 10^6$ CFU/7.0 µL) for 10 s, 30 s, 1 min, 3 min and 5 min in 1.5 ml centrifuge

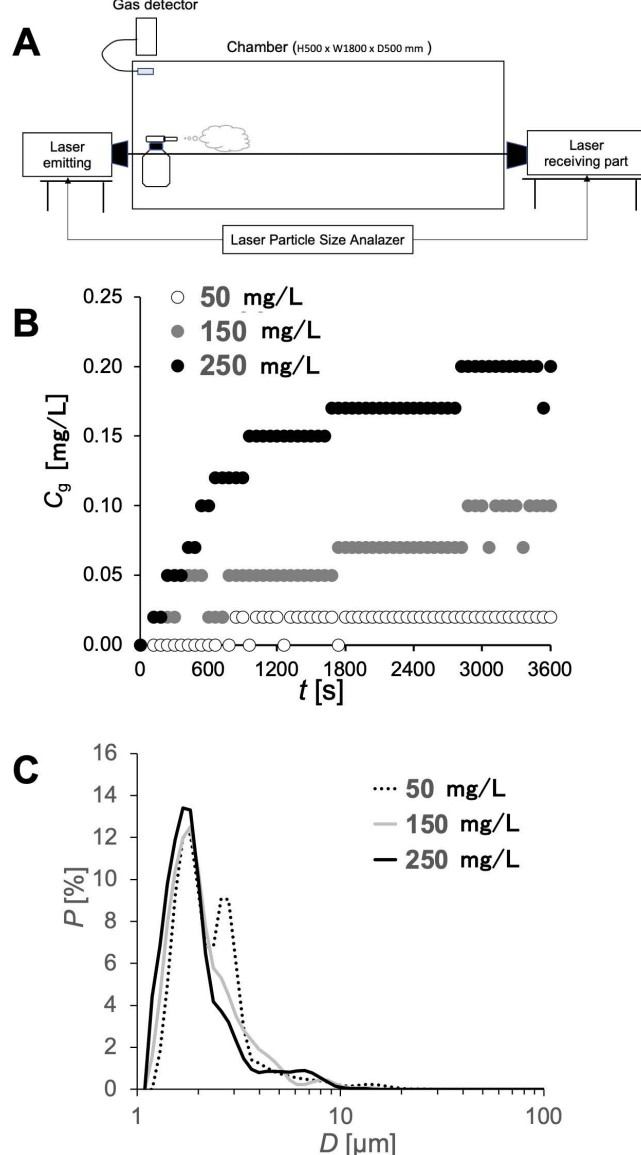

**Fig 1. Slightly acidic electrolyzed water (SAEW; iPOSH) particle size distribution and chlorine concentration in the SAEW aerosol disinfection chamber. (A)** Schematic diagram showing the setup of the chamber, chlorine concentration sensor, and laser measurement device during measurements. Rats were housed inside this chamber for a fixed period during aerosol disinfection exposure experiments. (B) Changes in chlorine concentration inside the chamber after the start of aerosol disinfection of slightly acidic electrolyzed water (SAEW). SAEW with chlorine concentrations of 50, 150, and 250 mg/L were sprayed into the chamber, and the chlorine concentration was monitored over time. **(C)** Particle size distribution of SAEW mist inside the chamber 1 h after the start of aerosol disinfection. SAEW with chlorine concentrations of 50, 150, and 250 mg/L was sprayed continuously for 1 h until the mist was stably distributed, and the particle size distribution of the mist was measured.

tubes at room temperature, followed by the addition of 35 μL of 0.1 M sodium thiosulfate to stop the reaction. For the negative control, PBS solution was mixed with the *E. coli* suspension. Notably, the mixed solutions containing *E. coli* were serially diluted with LB medium. Thereafter, the *E. coli* suspensions (100 CFU/200 μL) were spread on two LB agar plates and incubated at 37 °C for 48 h. Finally, the number of colonies was counted, and the sterilization effect of SAEW was evaluated.

## Assessment of the sterilization efficacy of SAEW via the aerosol disinfection method

Briefly, the sterilization efficacy of SAEW via aerosol disinfection was estimated using a spraying apparatus (NB-2N, SHIBATA SCIENTIFIC TECHNOLOGY Ltd., Tokyo, Japan) in the chamber (CLEA Japan, Inc., Tokyo, Japan). Aerosol disinfection of SAEW (15 ml) solution lasted for 1 h. Additionally, the sterilization effect of SAEW on *E. coli* was investigated using LB agar plate and absorbent cotton (Cut Cotton, Iwatsuki Co., Ltd., Tokyo, Japan).

Before treatment, *E. coli* suspension (100 CFU/200 μL) was spread on two LB agar plates. The plates were then placed in the center position opposite the spraying apparatus in the chamber (Fig 1A). After spraying the agar plates with SAEW (50 and 250 mg/L) for 1 h, 100 μL of 0.1 M sodium thiosulfate was added to the treated plates to terminate the reaction. For the negative control, PBS solution was sprayed. Thereafter, the plates were incubated at 37 °C for 48 h. Finally, the number of colonies formed was counted, and the sterilization effect of SAEW was evaluated.

Furthermore, absorbent cotton (20 mm × 20 mm × 5 mm) dabbed with 100 μL of *E. coli* suspension ($1.0 \times 10^6$ CFU/100 μL) was placed in the same positions described above in the chamber, followed by spatial application of SAEW (250 mg/L) for 1 h. For the negative control, untreated absorbent cotton dabbed with 100 μL of *E. coli* suspension was placed outside of the chamber for 1 h. Treated absorbent cotton with *E. coli* was transferred to 10 ml of LB medium in a 50 ml centrifuge tube and mixed by vortexing. *E. coli* suspension (100 CFU/200 μL) from the mixed solution was spread on two LB agar plates and incubated at 37 °C for 48 h. Thereafter, the number of colonies formed was counted, and the sterilization effect of SAEW was evaluated.

## Safety confirmation of aerial sprays of HClO solution in rats

To confirm safety to living organisms, rats were exposed to spatial sprays of 250 mg/L SAEW at 15 mL/h for 6 h a day, 5 days a week, for 3 months (65 days out of 90 days). Aerosol disinfection of SAEW was induced in a dedicated chamber for 1 h until sufficient diffusion was achieved. Thereafter, a cage containing a rat was placed in the chamber, and aerosol disinfection was continued for 6 h. During spraying, the airborne concentration of HClO in the chamber reached a steady-state level of approximately 0.20 mg/L, and the particle size distribution of the mist peaked at ~1.6 μm. To evaluate safety, the body weights, feed intake, and water intake of the rats were measured weekly. At 65 days of SAEW spatial application, urine samples were collected from the rats for biochemical analysis using a rat metabolic cage. At the end of the study, animals were anesthetized by administering a combination of medetomidine hydrochloride (0.15 mg/kg), midazolam (2 mg/kg), and butorphanol tartrate (2.5 mg/kg), and blood samples were collected. Animals were euthanized by exsanguination via incision of the vena cava while under deep anesthesia. Thereafter, the levels of inflammatory markers, including white blood cells and other components, in the blood and bronchoalveolar lavage fluid (BALF) were determined using flow cytometry (ADVIA 2120i; Siemens Healthcare Diagnostics Inc., Tarrytown, NY, USA). Additionally, changes in serum parameters were determined using the Hitachi Biochemistry Automated Analyzer Model 7180. Moreover, serum and blood cortisol concentrations were measured using the chemiluminescent immunoassay (CLEIA) method (IMMULITE 2000; Siemens Healthcare Diagnostics Products Ltd., Tarrytown, NY, USA). Furthermore, specific organs, including the eyes, nasal cavity, pharynx, lungs, liver, kidneys, adrenal glands, heart, spleen, stomach, esophagus, gonads (testes and ovaries), and intestines, were fixed in 10% buffered formalin and stained with hematoxylin and eosin, following standard procedures. Thereafter, the stained sections were viewed under a light microscope to assess histopathological changes and the presence of abnormalities or inflammation. Necropsy was conducted under veterinary supervision by Y.F. to ensure appropriate pathological assessment.

A total of 24 rats (12 males and 12 females) were allocated into two groups: an untreated control group and an SAEW aerosol disinfection group (n = 6 males and 6 females per group), with animals randomly assigned to each group. All assessments, including blood tests, blood biochemical analyses, bronchoalveolar lavage fluid (BALF) analysis, urine tests, and histological evaluations, were performed by an external service provider who was blinded to the experimental groups and unaware of the treatment conditions, thereby minimizing observer bias.

## Statistical analysis

All experiments were repeated at least three times. Significant differences between groups were determined using a two-tailed Student's *t*-test or one-way ANOVA with Tukey's multiple comparison test. All statistical analyses were performed using GraphPad InStat software (version 3.02) with the Tukey–Kramer Multiple Comparison test. Statistical significance was set at $p < 0.05$.

## Results

### Assessment of SAEW spatial distribution using laser measurement

HClO solutions at three concentrations ($C_l = 50$, 150, and 250 mg/L) were sprayed into the chamber using a mist generator at a flow rate of 15 mL/h. Thereafter, the spatial distribution of the mist was evaluated by measuring both the airborne concentration of HClO and the particle size distribution. Fig 1B shows the time course of HClO concentration in the chamber, with the horizontal axis representing time t [s] and the vertical axis representing gas-phase chlorine concentration Cg [mg/L]. Notably, the chlorine gas detector responds to chlorine derived from hypochlorous acid vaporized into the air. For $C_l = 50$ mg/L, the concentration Cg reached a plateau value of 0.02 at $t \sim 800$, indicating equilibrium between mist generation and volatilization. Similarly, higher solution concentrations led to higher equilibrium gas concentrations: $C_g = 0.10$ for $C_l = 150$, and $C_g = 0.20$ for $C_l = 250$. In all cases, the airborne concentration stabilized within 3000 s of mist generation.

Additionally, the particle size distribution of HClO mist in the chamber after 3000 s was measured. Fig 1C shows the size distribution, with the horizontal axis representing the particle diameter $D$ [μm] and the vertical axis representing the percentage $P$ [%] of the particles. Notably, $P$ is defined as the percentage of particles with a specific diameter relative to the total particles. Additionally, the particle size distribution results shown here are arithmetic averages of measurement results obtained for $3000 < t < 3600$. Considering that the mist generator used in this experiment supplies mist particles with an average particle size of 2–5 μm regardless of the Cl concentration, the measured particle size distribution peaked at a particle size of approximately $D = 1.6$. Generally, the particle size remained as small as the original size released by the mist generator, even after the spraying time had elapsed. In contrast, the HClO solution in the liquid volatilized from the mist into the surrounding air during introduction into the chamber. Notably, the volatilization process was characterized by a decrease in the volume of the mist and the particle size. Additionally, the higher the concentration of Cl in the disinfectant, the higher the volatilization process. Therefore, the distribution near particle size $D = 1.6$ for $C_l = 250$ exhibits larger values than in other cases. Eventually, the HClO concentrations in the air and liquid are expected to reach equilibrium, with the particle size converging to a constant value. Based on the time series of the concentration $C_g$ in the air (Fig 2), it can be confirmed that the solution with the highest HClO concentration ($C_l = 250$) showed the highest HClO volatilization into the surrounding environment.

Overall, the higher the concentration of HClO in the solution, the higher the volatilization from the mist, resulting in a higher concentration of HClO in the air. Notably, the concentration in the air became constant approximately 1 h after spraying. Additionally, the particle size of the sprayed mist remained small even after the spraying time had elapsed. Particularly, volatilization shifted the mean particle size distribution slightly to a smaller particle size than that of the mist generated by the mist generator.

### Sterilization effect of the SAEW solution against *E. coli* via the direct contact method

To evaluate the sterilization ability of SAEW, *E. coli* ($1.0 \times 10^6$ CFU) was incubated with 31.3, 62.5, 125, and 250 mg/L SAEW for 10 s, 30 s, 1 min, 3 min, and 5 min. After serial dilution, approximately 100 CFU of *E. coli* were cultured on LB agar plates for 48 h, after which the number of colonies was counted. In the control group, *E. coli* ($1.0 \times 10^6$ CFU) was incubated with PBS for 5 min. Treatment with all concentrations of SAEW inhibited the formation of *E. coli* colonies at all time points, with almost no colonies detected in the treatment groups (Fig 2A). In contrast, *E. coli* colony formation rate

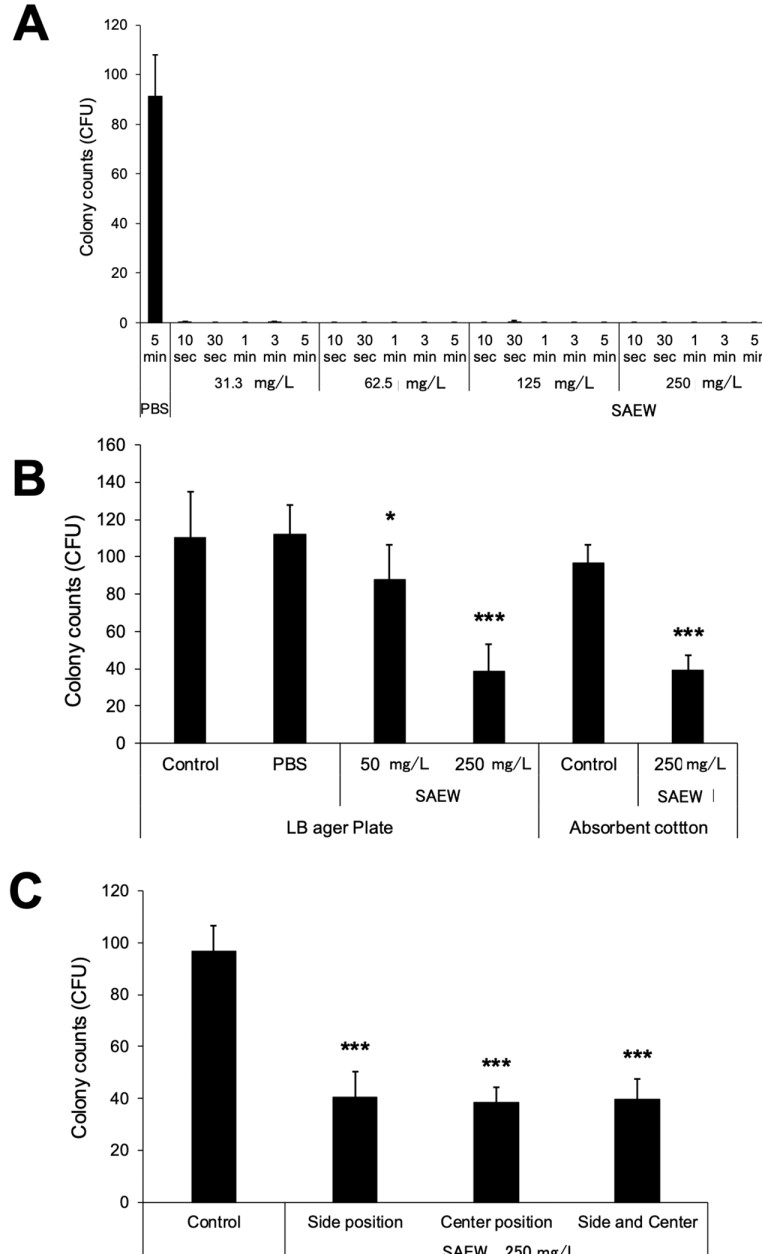

**Fig 2. Sterilization effect on *Escherichia coli* by direct addition or aerosol disinfection of SAEW (iPOSH). (A)** Sterilization effect of SAEW on *E. coli* via the direct contact method. *E. coli* (1.0×10⁶ colony formation unit [CFU]) were reacted directly with varying concentrations of SAEW (31.3, 62.5, 125, and 250 mg/L) for 10 s, 30 s, 1 min, 3 min, and 5 min. After reaction, the medium was serially diluted, and approximately 100 CFU *E. coli* were cultured on an LB agar plate, and the number of colonies was counted. Experiments were repeated at least three times. Each experiment used an independently prepared frozen aliquot of the bacterial stock. The bar graph indicates mean±standard deviation (SD). **(B)** Sterilization efficacy of SAEW against *E. coli* on an LB agar plate or the absorbent cotton via the aerosol disinfection method. *E. coli* (100 CFU) on an LB agar plate was exposed to varying concentrations (50 and 250 mg/L) of spatial sprays of SAEW for 1 **h.** After reaction, *E. coli* on an LB agar plate were cultured, and the number of colonies were counted. Experiments were repeated at least three times. Each experiment used an independently prepared frozen aliquot of the bacterial stock. The bar graph indicates mean±standard deviation (SD). Compared with the no-spray control, * indicates p<0.05 and *** indicates p<0.001, as determined using one-way ANOVA followed by Tukey's multiple comparison test. **(C)** Sterilization efficacy of SAEW against *E. coli* on the absorbent cotton following aerosol disinfection. *E. coli* (1.0×10⁶ CFU) on the absorbent cotton were exposed to spatial sprays of SAEW (250 mg/L) for 1 h in the chamber. The absorbent cotton with *E. coli* was placed in center and opposite side positions in the chamber. After reaction, *E. coli* on the absorbent cotton was suspended in LB medium. *E. coli* suspension was serially diluted, and approximately 100 CFU of *E. coli* were cultured on the LB agar plate,

and the number of colonies was counted. Experiments were repeated at least three times. Each experiment used an independently prepared frozen aliquot of the bacterial stock. The bar graph indicates mean ± standard deviation (SD). Compared with the no-spray control, *** indicates p < 0.001, as determined using one-way ANOVA followed by Tukey's multiple comparison test.

was 91.4 ± 16.7% (mean ± SD) in the PBS (control) group. Overall, these results indicate that treatment with SAEW (31.3 mg/L) for 10 s effectively sterilizes *E. coli* (1.0 × 10⁶ CFU). Importantly, these bactericidal effects were consistent with those observed in a previous study targeting coronaviruses using SAEW [13].

## Sterilization effect of SAEW via aerosol disinfection

In this study, we examined the sterilization efficacy of SAEW against *E. coli* following aerosol disinfection. An agar plate experiment was performed to assess the sterilization effects of SAEW against *E. coli* in LB medium. Specifically, 100 CFU of *E. coli* were spread on the LB agar plate and treated with 50 and 250 mg/L SAEW and PBS (negative control). SAEW treatment significantly inhibited colony formation in a concentration-dependent manner (88.0 ± 18.3, 38.6 ± 14.7, and 112.3 ± 15.5% in the 50 mg/L, 250 mg/L, and PBS groups, respectively; Fig 2B, LB agar plate panel). Additionally, a higher number of colonies were formed around the boundaries (sides 44.0 ± 15.7%) of the chamber than in the center position (27.2 ± 13.5%) in the 250 mg/L group, but not in the 50 mg/L group. Collectively, these data indicate that the position of LB agar plates in the chamber affects the sterilization efficiency of SAEW.

Additionally, we examined the sterilization ability of SAEW against *E coli* on absorbent cotton. Specifically, absorbent cotton was dabbed in the *E. coli* (1.0 × 10⁶ CFU) and treated with 250 mg/L SAEW. After treatment, *E. coli* on the absorbent cotton was suspended in LB medium and serially diluted. Approximately 100 CFU of *E. coli* were spread on the LB agar plate and cultured. Compared with that in the control group (96.7 ± 9.8%), treatment with 250 mg/L SAEW significantly decreased *E. coli* colony formation to 39.5 ± 7.8% (Fig 2B, Absorbent cotton panel). Notably, the number of colonies formed was comparable between the chamber boundaries (side) and center position (Fig 2C). Collectively, these results indicate that aerosol disinfection of SAEW was partially effective in eradicating *E. coli* on LB agar plate and absorbent cotton.

## Evaluation of the safety of aerial sprays of SAEW in rats

Spatial/aerial spraying of the HClO solution did not induce any morphological or behavioral abnormalities in exposed rats. Additionally, there was no significant difference in body weight between the experimental and control groups (males, $p = 0.454$; females, $p = 0.591$; Table 1). Importantly, body weight, food intake, and water intake remained within the normal range when compared to historical reference data for CD(SD) IGS rats [15]. Additionally, we assessed hematological parameters of the rat after 65 days of spatial application of SAEW and found that there were no significant differences red blood cell counts (males, $p = 0.456$; females, $p = 0.275$) and hemoglobin levels (males, $p = 0.279$; females, $p = 0.205$), indicators of respiratory function and oxygen transport, between the groups. Moreover, white blood cell (males, $p = 0.196$; females, $p = 0.069$), lymphocyte (males, $p = 0.702$; females, $p = 0.464$), neutrophil (males, $p = 0.753$; females, $p = 0.697$), and platelet (males, $p = 0.316$; females, $p = 0.251$) counts were within normal range, confirming the absence of any inflammatory response.

Moreover, blood biochemical tests were conducted to evaluate organ function. Notably, there were no significant differences ($p < 0.05$) in the levels of the liver function markers alanine transaminase (ALT), aspartate aminotransferase (AST), and total bilirubin (T-BIL) between the experimental and control groups, confirming normal hepatic function. Similarly, there were no significant differences ($p < 0.05$) in the levels of the renal function markers blood urea nitrogen (BUN) and creatinine (CRE) between the groups, indicating no kidney abnormalities. Electrolyte levels (Na, K, and Cl) were within the normal range, with no disturbances observed. Additionally, there were no significant differences ($p > 0.05$) in blood glucose

**Table 1. Physiological, hematological, and biochemical parameters in rats under SAEW spraying conditions.**

| | | Control | Spatial spray of SAEW | P value |
|---|---|---|---|---|
| Biological Indices | | | | |
| Body weight (g) | Male | 588±67 | 615±39 | 0.454 |
| (at 21 weeks of age after aerosol disinfection) | Female | 296±7 | 291±20 | 0.591 |
| Weekly water intake (g) | Male | 36.9±2.1 | 45.9±5.1 | 0.118 |
| (at week 13 of aerosol disinfection) | Female | 24.4±1.5 | 23.9±2.4 | 0.805 |
| Weekly food intake (g) | Male | 28.7±2.7 | 30.3±1.9 | 0.533 |
| (at week 13 of aerosol disinfection) | Female | 18.2±0.7 | 17.5±0.7 | 0.372 |
| Complete Blood Count | | | | |
| red blood cell count (x 10⁴/µL) | Male | 843±31 | 874±80 | 0.456 |
| | Female | 731±42 | 765±50 | 0.275 |
| Hemoglobin (g/dL) | Male | 14.9±0.5 | 15.6±1.2 | 0.279 |
| | Female | 13.8±0.7 | 14.4±0.8 | 0.205 |
| white blood cell count (/µL) | Male | 9133±1862 | 7467±1945 | 0.196 |
| | Female | 4100±924 | 5833±1601 | 0.069 |
| lymphocytes (%) | Male | 75.2±6.1 | 73.9±3.8 | 0.702 |
| | Female | 81.9±2.8 | 83.2±2.6 | 0.464 |
| neutrophil (%) | Male | 20.8±5.7 | 21.7±3.4 | 0.753 |
| | Female | 14.2±2.0 | 13.6±2.5 | 0.697 |
| platelet (x10⁴/uL) | Male | 106.1±6.3 | 99.1±13.2 | 0.316 |
| | Female | 103.0±17.8 | 92.5±4.9 | 0.251 |
| Serum Biochemistry | | | | |
| Alanine Aminotransferase, ALT (IU/L) | Male | 26±2 | 27±5 | 0.715 |
| | Female | 19±3 | 19±3 | 0.932 |
| Aspartate Aminotransferase, AST (IU/L) | Male | 71±15 | 65±9 | 0.438 |
| | Female | 51±4 | 57±8 | 0.175 |
| T-BIL (mg/dL) | Male | 0.07±0.01 | 0.06±0.01 | 0.828 |
| | Female | 0.08±0.01 | 0.11±0.02 | 0.039 |
| BUN (mg/dL) | Male | 12.0±1.5 | 13.9±2.1 | 0.130 |
| | Female | 12.3±0.8 | 11.9±1.4 | 0.605 |
| CRE (mg/dL) | Male | 0.31±0.04 | 0.34±0.10 | 0.572 |
| | Female | 0.31±0.03 | 0.32±0.05 | 0.73 |
| ALB (g/ dL) | Male | 3.8±0.1 | 3.5±0.3 | 0.045 |
| | Female | 4.3±0.3 | 4.1±0.4 | 0.376 |
| Na (mEq/L) | Male | 139±2 | 137±1 | 0.088 |
| | Female | 137±1 | 136±2 | 0.181 |
| K (mEq/L) | Male | 3.2±0.3 | 3.9±0.5 | 0.019 |
| | Female | 3.1±0.1 | 3.1±0.4 | 0.715 |
| Cl (mEq/L) | Male | 96±1 | 97±2 | 0.869 |
| | Female | 99±2 | 97±2 | 0.256 |
| Glu (mg/dL) | Male | 206±42 | 249±37 | 0.114 |
| | Female | 184±21 | 159±21 | 0.094 |
| cortisol (ug/dL) | Male | 0.3±0.1 | 0.2±0.0 | 0.220 |
| | Female | 0.4±0.1 | 0.3±0.1 | 0.132 |
| BALF Cell Count | | | | |
| red blood cell count (x 10⁴/µL) | Male | 2±2 | 1±0 | 0.466 |
| | Female | 1±0 | 1±0 | 1.000 |

*(Continued)*

**Table 1.** (Continued)

|  |  | Control | Spatial spray of SAEW | P value |
|---|---|---|---|---|
| Hemoglobin (g/dL) | Male | 0.1±0 | 0.1±0 | 1.000 |
|  | Female | 0.1±0 | 0.1±0 | 1 |
| white blood cell count (/μL) | Male | 317±134 | 183±69 | 0.086 |
|  | Female | 183±107 | 200±100 | 0.804 |
| lymphocytes (%) | Male | 80.8±8.2 | 78.0±7.3 | 0.580 |
|  | Female | 75.6±15.1 | 81.1±7.6 | 0.484 |
| neutrophil (%) | Male | 14.0±7.3 | 15.2±6.6 | 0.790 |
|  | Female | 15.2±8.9 | 12.7±4.8 | 0.594 |
| platelet (x10⁴/uL) | Male | 2.3±0.7 | 1.8±0.7 | 0.268 |
|  | Female | 2.2±0.3 | 2.3±0.5 | 0.799 |

(GLU) and cortisol (an indicator of stress response) levels between the experimental and control groups. Although there were significant differences (p<0.05) in potassium (K) and albumin (ALB) levels between the groups (K: control 3.2±0.3, SAEW 3.9±0.5, p=0.019; ALB: control 3.8±0.1, SAEW 3.5±0.3, p=0.045), these differences were observed only in males, whereas females showed no significant differences, suggesting statistical variation without biological or clinical relevance. While the values differ somewhat from the historical reference data reported for 30-week-old CD(SD) IGS rats by Charles River Japan (K: 4.5±0.4 mmol/L, ALB: 4.0±0.2 g/dL, n=9) [15], they are not indicative of any overt adverse effects. Overall, the results do not suggest that SAEW aerosol disinfection had biologically meaningful impacts on these parameters.

Histological examination showed that there were no inflammatory changes in organs directly exposed to SAEW, including the eye, nasal cavity, pharynx, and lungs (Figs 3A–3H). Although minor inflammation was noted in the lungs of one male control rat (not exposed to SAEW aerosol disinfection) (Fig. 3G2), this was considered an individual variation. Additionally, BALF analysis confirmed no inflammatory response, as evidenced by similar levels of white blood cells, neutrophils, and lymphocytes between the experimental and control groups (*p*>0.05). Platelet counts, indicative of bleeding tendencies or coagulopathy in the lungs, were within the normal range (*p*>0.05).

Urinalysis showed no presence of glucose or occult blood, confirming the absence of renal dysfunction. Collectively, these results demonstrate that exposure to spatial application of SAEW for 90 days does not induce adverse physiological, hematological, and histological effects in rats, supporting its safety for environmental disinfection applications.

## Discussion

In this study, we confirmed both the bactericidal effectiveness and safety of SAEW administered via aerosol disinfection. Although assessing higher concentrations may be of interest, 250 mg/L represents a relatively high level, and evaluating higher doses falls outside the scope of the present study. To our knowledge, this is the first investigation to demonstrate the safety of repeated SAEW exposure in an animal model. Despite using a relatively high concentration (250 mg/L), no adverse effects were observed in rats, supporting previous findings showing the safety of orally administered low-concentration SAEW (5 mg/L) in mice [16]. SAEW effectively inactivated airborne and droplet-associated pathogens, underscoring its potential as a frontline disinfection strategy against respiratory viruses such as SARS-CoV-2 and influenza virus.

We also evaluated the bactericidal properties of SAEW against *E. coli*, a gram negative bacterium known for its resistance to disinfectants. Because enveloped viruses such as SARS-CoV-2 readily lose infectivity upon disruption of their lipid membrane, they are generally more susceptible to disinfectants than *E. coli*. Our previous study demonstrated that SAEW (≥ 56.5 mg/L) inactivated coronaviruses within 10 s, decreasing viral titers from 7.0 log TCID$_{50}$/mL to below the detection limit (≤ 1.5 log TCID$_{50}$/mL) [13]. Consistent with this, the present study showed that SAEW aerosolization

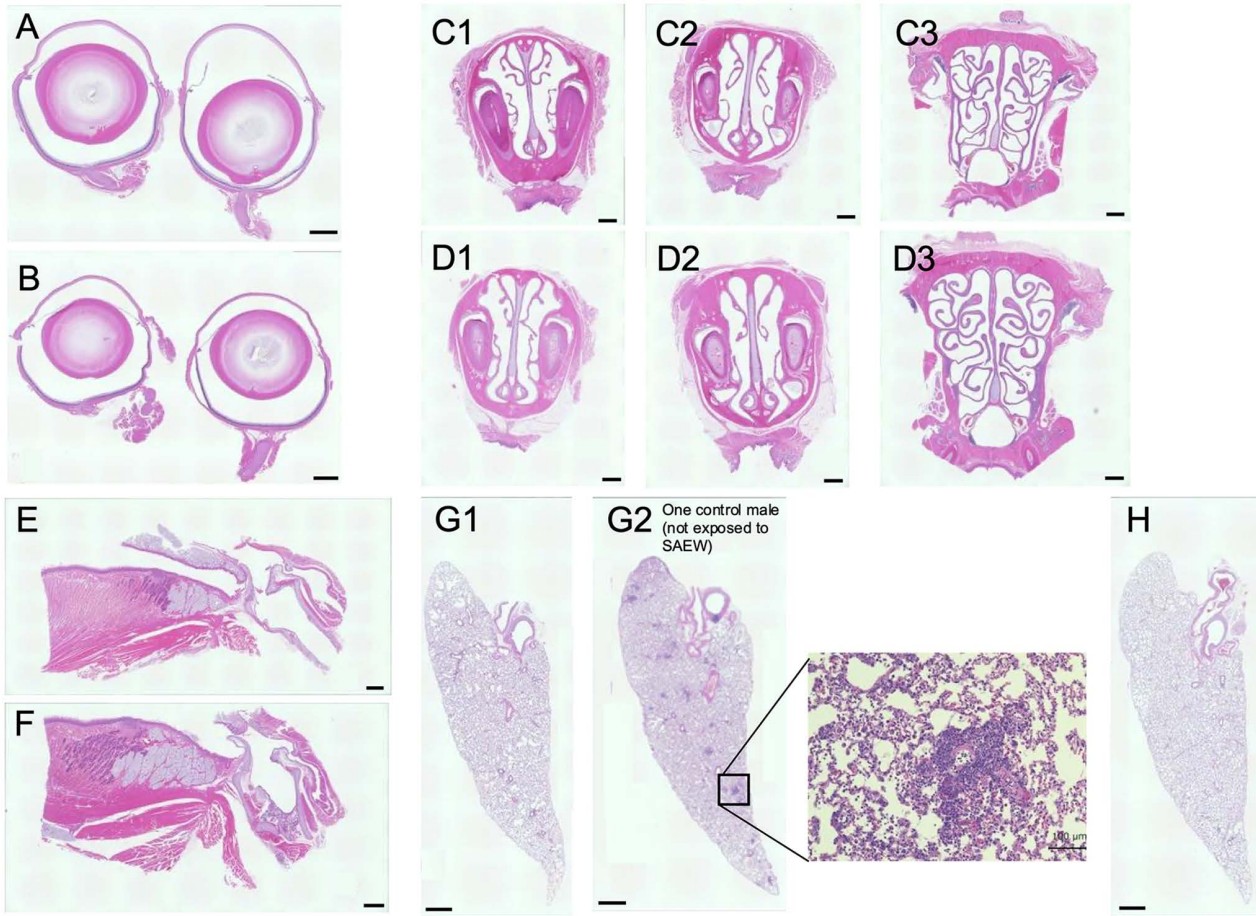

**Fig 3. Histological sections of rat tissues exposed to aerosol disinfection of SAEW.** Histological sections of the eye (A: control, B: SAEW aerosol disinfection), nasal cavity (C: control, D: SAEW aerosol disinfection) at the anterior, middle, and posterior regions, pharynx (E: control, F: SAEW aerosol disinfection), and lungs (G: control, H: SAEW aerosol disinfection) of rats exposed to SAEW. Representative images at ×4 magnification showing hematoxylin and eosin (H&E) staining. G2 represents a male rat that was not exposed to SAEW aerosol disinfection but exhibited mild pulmonary inflammation. Scale bar: 1 mm.

effectively eliminated *E. coli* from LB agar plates and absorbent cotton, indicating its ability to inactivate both airborne and surface-adhered microorganisms.

SAEW has been widely reported as an effective antimicrobial agent with broad-spectrum activity. Previous studies documented its bactericidal efficacy against *Bacillus cereus*, *B. subtilis*, *Pseudomonas aeruginosa*, *Salmonella enterica*, *Staphylococcus aureus*, *Cladosporium*, and *S. epidermidis* [17]. Comparative work also demonstrated stronger bactericidal activity than sodium hypochlorite at equivalent chlorine levels and effective inactivation of *E. coli*, *P. aeruginosa*, and *S. epidermidis* in both solution and spray forms [18]. These results align with our findings and support the broad antimicrobial potential of SAEW. Additionally, SAEW spraying markedly reduced microbial contamination in municipal solid-waste sorting rooms (15.7–98.0% reduction) [19], highlighting the practical applicability of SAEW aerosolization. Although SAEW aerosolization proved effective against *E. coli* in the present study, further research is necessary to evaluate its efficacy against diverse bacterial and viral pathogens in aerosolized form.

SAEW also has potential applications in livestock hygiene, food safety, and agriculture. The livestock industry faces threats from pathogens such as PRRSV, FMDV, and highly pathogenic avian influenza virus (AIV) [20]. In the food

industry, Norovirus and *Salmonella* spp. remain major concerns, while in agriculture, plant pathogens such as ToMV, PMMV, and *Botrytis cinerea* negatively impact crop yields. SAEW aerosolization may offer a non-invasive and rapid disinfection method in these fields. Its bactericidal activity has been demonstrated against various bacterial species, including *Listeria* spp. [21,22], *Salmonella* spp. [23], *E. coli* [23], coliforms [24,25], and *Bacillus* spp. [26], and SAEW exhibits potent antiviral effects against AIV [27].

The National Institute of Technology and Evaluation (NITE) in Japan has affirmed that HClO solutions ≥ 35 mg/L exhibit strong disinfectant activity against SARS-CoV-2. However, conventional electrolysis-generated HClO solutions often have low available chlorine (< 60 mg/L) and poor stability. In this study, we used iPOSH (Local Power Co., Ltd., Akita, Japan), a highly purified hypochlorous acid solution produced through ion-exchange purification of sodium hypochlorite as a form of SAEW. This method enables the stable production of HClO solutions up to 1,000 mg/L. These solutions remained stable for over a year under room-temperature, dark storage, with a 250-mg/L preparation retaining 93.6% (238 mg/L) of its initial concentration after six months and 83.4% (212 mg/L) after one year (S1 Fig.). Compared with alcohol-based disinfectants and sodium hypochlorite, HClO offers advantages including low toxicity, minimal irritation, and non-flammability, making it suitable for applications such as hand sanitization and ophthalmic and dental sprays [28].

Although SAEW is generally considered low in toxicity, this study represents the first *in vivo* evaluation of repeated aerosol disinfection in animals. No adverse effects were observed, likely due to the rapid neutralization of SAEW upon contact with biological tissues. Future studies could assess higher exposure levels and extended durations to establish broader safety margins. While caution is required when extrapolating these findings to humans, particularly sensitive populations such as infants, elderly individuals, and those with respiratory conditions, the present results support the safe use of SAEW in animal models and suggest potential benefits in human environments when applied appropriately.

Overall, this study provides strong evidence for the bactericidal efficacy and safety of SAEW aerosolization. Our findings highlight the potential of SAEW as an innovative infection control strategy with applications across medical, livestock, food, and agricultural industries. Future research should focus on elucidating its virucidal effects under diverse environmental and real-world conditions.

## Supporting information

**S1 Fig. Changes in available chlorine concentration of iPOSH slightly acidic electrolyzed water (SAEW) stored at room temperature (approximately 20 °C). iPOSH SAEW with an initial available chlorine concentration of 250 mg/L was stored at room temperature for up to 24 months.** Data represent the mean ± standard error of three independent experiments. The standard errors were too small to be visible and are therefore obscured by the markers.
(TIFF)

## Author contributions

**Data curation:** Wataru Yamazaki.

**Investigation:** Takanori Oikawa, Kana Akinaga, Megumi Yanagida, Wataru Yamazaki, Yuriko Fujii, Misako Higashiya, Takahiro Adachi, Shinsuke Seki.

**Methodology:** Tomoko Sato, Yuki Mukoda, Koya Terada, Shinsuke Seki.

**Resources:** Tomoko Sato, Yuki Mukoda, Koya Terada.

**Supervision:** Koya Terada, Shinji Yamasaki, Takahiro Adachi, Shinsuke Seki.

**Visualization:** Kana Akinaga, Wataru Yamazaki, Shinsuke Seki.

**Writing – original draft:** Wataru Yamazaki, Takahiro Adachi, Shinsuke Seki.

**Writing – review & editing:** Shinji Yamasaki, Shinsuke Seki.

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
