## [Decision Letter · Decision Letter 0]

18 Nov 2025

Dear Dr. Seki,

Thank you for submitting your manuscript to PLOS ONE. After careful consideration, we feel that it has merit but does not fully meet PLOS ONE’s publication criteria as it currently stands. Therefore, we invite you to submit a revised version of the manuscript that addresses the points raised during the review process.

We look forward to receiving your revised manuscript.

Kind regards,

Jiafu Li

Academic Editor

PLOS ONE

Journal Requirements:

2. To comply with PLOS One submissions requirements, in your Methods section, please provide additional information regarding the experiments involving animals and ensure you have included details on (1) methods of sacrifice, (2) methods of anesthesia and/or analgesia, and (3) efforts to alleviate suffering.

This study was conducted as a collaborative research project with Local Power Co., Ltd., and was financially supported by the company. Although some authors (T.S., Y.M., and K.T.) are employees of Local Power Co., Ltd., their primary role was to provide SAEW (slightly acidic electrolyzed water), and the evaluation of its effects was carried out entirely by university staff. All experiments and analyses were conducted objectively and scientifically to ensure the integrity and fairness of the research.

5. Please include a copy of Table 1, which you refer to in your text on page 17.

Reviewers' comments:

Reviewer's Responses to Questions

**Comments to the Author**

1. Is the manuscript technically sound, and do the data support the conclusions?

Reviewer #1: Yes

Reviewer #2: Yes

Reviewer #3: Yes

Reviewer #4: Yes

2. Has the statistical analysis been performed appropriately and rigorously?

Reviewer #1: Yes

Reviewer #2: Yes

Reviewer #3: No

Reviewer #4: I Don't Know

3. Have the authors made all data underlying the findings in their manuscript fully available?

Reviewer #1: Yes

Reviewer #2: Yes

Reviewer #3: No

Reviewer #4: Yes

4. Is the manuscript presented in an intelligible fashion and written in standard English?

Reviewer #1: Yes

Reviewer #2: Yes

Reviewer #3: Yes

Reviewer #4: Yes

Reviewer #1: This study evaluates the efficacy and safety of SAEW applied through spatial spraying, combining aerosol characterization, bactericidal testing, and long-term inhalation exposure in rats. The work is well executed and provides consistent evidence supporting the safety of SAEW as a spatial disinfectant.

1. The Introduction should be expanded to clarify the rationale for conducting in vivo experiments in rats. The authors are encouraged to explain why animal testing is necessary, how it addresses systemic or cumulative effects not captured by in vitro data, and how it links laboratory findings to real-world safety evaluation.

2.The description of the experimental design could be made more transparent. The authors are advised to specify the number of animals in each group and clarify whether randomization or blinding was applied during data collection. Providing supplementary materials with additional data summaries or relevant methodological details would allow readers to better assess the reproducibility and completeness of the study.

3.In the Discussion, I recommend that the authors further elaborate on the study’s limitations and outline directions for future research. If possible, they may also briefly consider potential biological explanations for the absence of adverse findings.

Reviewer #2: Major Concerns:

1. The study lacks a detailed discussion on the potential risks of chronic exposure in humans, particularly in sensitive populations.

2. Although SAEW was found safe in rats, extrapolation to human environments (e.g., schools, hospitals) warrants further discussion.

3. The experimental design should mention whether blinding was used during histopathological assessments to minimize observer bias.

Minor Issues:

1. The graphical representation of data (e.g., colony counts, particle size distribution) could be improved for clarity.

2. Some parts of the text are overly descriptive and could be streamlined to maintain scientific focus.

3. The potential for formation of disinfection byproducts under various environmental conditions should be acknowledged.

Reviewer #3: This manuscript investigates the chronic safety of slightly acidic electrolyzed water (SAEW) applied as an aerosol (“spatial spraying”) in Sprague–Dawley rats over a 3-month period. The study evaluates hematological, biochemical, and histopathological parameters to determine whether repeated SAEW exposure induces any toxicological or inflammatory changes.

The topic is relevant to environmental disinfection and public health, particularly in the context of widespread use of SAEW as a disinfectant. The study is generally well organized, and the results are clearly presented. However, several critical methodological details are missing, and the experimental design would benefit from additional rigor and transparency.

Major Comments

1. Missing sample size (n) information

The manuscript does not report the sample size (n) for any experiment.

In the animal exposure study, the number of rats per group (male and female) is not specified in the Methods, Results, or Table 1, making it impossible to assess statistical power or variability.

Similarly, in the bactericidal assays (Figure 2), the statement “each experiment was repeated at least three times” does not clarify whether these were technical or biological replicates.

None of the figure legends indicate the value of n or whether the error bars represent SD or SEM.

Please provide this information for all datasets to ensure transparency and reproducibility in line with PLOS reporting standards.

2. Insufficient exposure characterization

The description of the exposure system in the Methods (“Safety confirmation of aerial sprays of HClO solution in rats”) lacks the quantitative detail needed to assess the actual inhalation dose.

The authors report that rats were exposed to 250 ppm HClO solution at 15 mL h⁻¹ for 6 h day⁻¹, 5 days week⁻¹ for 3 months, but this only specifies the liquid concentration, not the actual airborne chlorine concentration or environmental conditions within the chamber.

Please provide quantitative or literature-based estimates of airborne HClO levels, along with any information about chamber temperature, humidity, or airflow, to clarify the true exposure conditions.

3. Inconsistent reporting of albumin data and lack of reference ranges

In the Results section, the authors mention that both potassium and albumin levels showed statistically significant differences (p < 0.05) but were “within physiological range.”

However, albumin data are not presented in Table 1 or any figure, and no reference ranges are provided to justify this interpretation.

Please include the albumin data (mean ± SD and p-value) in Table 1 and provide appropriate reference ranges for CD(SD) IGS rats (either from internal data or published literature) to support the statement that all values were within normal limits.

This clarification is critical for consistency and for verifying the authors’ interpretation of biochemical results.

4.Ambiguous histopathology interpretation

The statement that “minor inflammation was noted in one male control rat, considered an individual variation” is qualitative and lacks supporting evidence.

Please clarify whether the histopathological evaluation was conducted in a blinded manner, and consider providing semi-quantitative scoring or representative histological images to substantiate this interpretation.

Including these details would improve objectivity and strengthen the reliability of the histopathological assessment.

Minor Comments

1. Ensure consistent unit formatting (μL vs µl, mg/dL vs mg dL⁻¹).

2. The Discussion repeats parts of the Introduction; it could be shortened.

3. Clearly define significance levels (*, **, ***) in all figures and legends.

Reviewer #4: Subject: Effective and Safe: Long-Term Spatial Spraying of Slightly Acidic Electrolyzed Water Causes No Harm in Rats

Manuscript Number: PONE-D-25-54069

Dear Editor,

Thank you for the opportunity to revise the manuscript. We would like to share the following issues noted during the revision process and seek your guidance or confirmation where needed:

Minor Comments:

• Abstract

1. Improve clarity in language by simplifying some sentences and ensuring consistent terminology (e.g., “spatial spraying” vs. “aerosol disinfection”).

2. Ensure units are consistently reported (e.g., mg/L vs. ppm for chlorine concentration).

• Results

The study is relevant and shows that SAEW is effective for spatial disinfection and appears safe in animal exposure tests. However, some clarifications are needed. Please provide more details on the spraying chamber conditions, include clearer statistical reporting, and briefly justify using only E. coli as the test organism. With these points addressed, the manuscript will be strengthened.

• Disscussion

The discussion clearly highlights the significance of SAEW in disinfection and its safety for long-term exposure. However, the section could be strengthened by briefly acknowledging limitations, such as the use of only E. coli as the test bacterium and the controlled laboratory conditions. It would also be helpful to emphasize the need for real-world field validation. Overall, the conclusions are reasonable and supported by the data.

• Reference Updates Needed:

Several references need to be updated or corrected with more recent and relevant citations.

**Do you want your identity to be public for this peer review?** For information about this choice, including consent withdrawal, please see our Privacy Policy

Reviewer #1: No

Reviewer #2: No

Reviewer #3: No

Reviewer #4: **Yes:**  Magda Elsayed Abd-Elgawad

---

## [Author Response · Author response to Decision Letter 1]

18 Dec 2025

Journal Requirements:

Answer 1: Thank you for your comment. We have revised the manuscript to comply with the PLOS ONE style requirements. Specifically, we inserted a table directly after the paragraph in which it is first cited (Lines 369-370) and placed each figure caption immediately after its first citation in the text (Fig. 1: Lines 125-136; Fig. 2: Lines 265-289; Fig. 3: Lines 372-378). We also updated the Funding section as requested (Lines 459-463).

2. To comply with PLOS One submissions requirements, in your Methods section, please provide additional information regarding the experiments involving animals and ensure you have included details on (1) methods of sacrifice, (2) methods of anesthesia and/or analgesia, and (3) efforts to alleviate suffering.

Answer 2: Thank you for your comment. We have revised the Methods section to include the required information on (1) methods of sacrifice, (2) methods of anesthesia and/or analgesia, and (3) efforts to alleviate suffering. Specifically, we added the following:

“At the end of the study, animals were anesthetized by administering a combination of medetomidine hydrochloride (0.15 mg/kg), midazolam (2 mg/kg), and butorphanol tartrate (2.5 mg/kg), and blood samples were collected. Animals were euthanized by exsanguination via incision of the vena cava while under deep anesthesia (Lines 194-198). All efforts were made to minimize animal suffering throughout the experiment. Animal handling and monitoring, including evaluation of activity, posture, and any abnormal signs, were performed under the supervision of Y.F., a licensed veterinarian, and T.O., a certified laboratory animal technologist instructor accredited by the Japanese Association for Laboratory Animal Science (JALAS). (Lines 93-97)”

This study was conducted as a collaborative research project with Local Power Co., Ltd., and was financially supported by the company. Although some authors (T.S., Y.M., and K.T.) are employees of Local Power Co., Ltd., their primary role was to provide SAEW (slightly acidic electrolyzed water), and the evaluation of its effects was carried out entirely by university staff. All experiments and analyses were conducted objectively and scientifically to ensure the integrity and fairness of the research.

Answer 3: Thank you for your comment. We have revised the Competing Interests statement to provide a clearer description of the role of Local Power Co., Ltd. in this study. Specifically, we added the following text to the manuscript: “This study was conducted as a collaborative research project with Local Power Co., Ltd., which provided financial support and supplied SAEW (slightly acidic electrolyzed water, iPOSH) via delivery. Although some authors (T.S., Y.M., and K.T.) are employees of Local Power Co., Ltd., their involvement was limited to performing pre-shipment quality confirmation of SAEW (including available chlorine concentration, pH, and basic performance checks). All animal experiments, performance testing of SAEW prior to use, and evaluation of its effects were conducted independently by university staff at Akita University. All experiments and analyses were carried out objectively and scientifically to ensure the integrity and fairness of the research.”Additionally, as requested, we have included the following statement to confirm compliance with PLOS ONE policies: “This does not alter our adherence to PLOS ONE policies on sharing data and materials. (Lines 448-457)” There are no restrictions on data or material sharing.

Answer 4: Thank you for your comment. We have removed the phrase “data not shown” and uploaded the corresponding stability data as a Supporting information (S1 File, Line 444-445). Accordingly, the sentence in the manuscript has been revised to cite this supplementary material. For instance, a 250 mg/L solution retained 93.6% (238 mg/L) of its initial concentration after six months and 83.4% (212 mg/L after one year (S1 File) (Line 424-426).

5. Please include a copy of Table 1, which you refer to in your text on page 17.

Answer 5: Thank you for your comment. We have included a copy of Table 1 as requested (Lines 369-370).

Answer 6: Thank you for your comment. We have added the suggested references and expanded the Discussion accordingly (Lines 402-407; References 18, 19).

In the submitted manuscript, changes made in response to editor and reviewer comments are highlighted in light blue. English language corrections made during professional editing are highlighted in yellow. Furthermore, in the Results section, we adjusted references to figures to clarify which figure supports each statement; these changes are highlighted in purple. This color-coding is intended to help the editor and reviewers quickly identify the different types of revisions made throughout the manuscript.

Response to Reviewers

Reviewer #1: This study evaluates the efficacy and safety of SAEW applied through spatial spraying, combining aerosol characterization, bactericidal testing, and long-term inhalation exposure in rats. The work is well executed and provides consistent evidence supporting the safety of SAEW as a spatial disinfectant.

1. The Introduction should be expanded to clarify the rationale for conducting in vivo experiments in rats. The authors are encouraged to explain why animal testing is necessary, how it addresses systemic or cumulative effects not captured by in vitro data, and how it links laboratory findings to real-world safety evaluation.

Answer Reviewer #1-1, Thank you for your valuable comment. We have revised the Introduction to clarify the rationale for conducting in vivo experiments. Specifically, we added a statement explaining that spatial spraying may lead to systemic or cumulative effects that cannot be adequately assessed using in vitro methods alone (Lines 78-80, “Because aerosol disinfection may exert systemic or cumulative effects not detected through in vitro assays, we performed an in vivo safety evaluation using rats, assessing physiological and histological parameters.”). This addition highlights the necessity of using a rat model to evaluate biological responses under conditions that more closely reflect real-world exposure.

2.The description of the experimental design could be made more transparent. The authors are advised to specify the number of animals in each group and clarify whether randomization or blinding was applied during data collection. Providing supplementary materials with additional data summaries or relevant methodological details would allow readers to better assess the reproducibility and completeness of the study.

Answer Reviewer #1-2, Thank you for your comment. We have clarified the experimental design in the Methods section. A total of 24 rats (12 males and 12 females) were allocated into two groups: an untreated control group and an iPOSH spatial spraying group (n = 6 males and 6 females per group), with animals randomly assigned to each group. All assessments, including blood tests, blood biochemical analyses, bronchoalveolar lavage fluid (BALF) analysis, urine tests, and histological evaluations, were performed by an external service provider who was blinded to the experimental groups and unaware of the treatment conditions, thereby minimizing observer bias (Methods, Line 211-216).

3.In the Discussion, I recommend that the authors further elaborate on the study’s limitations and outline directions for future research. If possible, they may also briefly consider potential biological explanations for the absence of adverse findings.

Answer Reviewer #1-3, Thank you for your valuable suggestion. We have revised the Discussion to clarify the novelty of this study as the first in vivo evaluation of repeated spatial spraying of slightly acidic electrolyzed water (SAEW) in living animals. We now note that no adverse effects were observed under the experimental conditions, likely due to the low toxicity and rapid neutralization of SAEW upon contact with biological tissues.

In addition, we addressed the study’s limitations and outlined future research directions, including the need for studies with higher exposure levels and longer durations. These revisions have been incorporated into the final paragraph of the Discussion (Lines 430–437), as follows:

“Although SAEW is generally considered low in toxicity, this study represents the first in vivo evaluation of repeated aerosol disinfection in animals. No adverse effects were observed, likely due to the rapid neutralization of SAEW upon contact with biological tissues. Future studies could assess higher exposure levels and extended durations to establish broader safety margins. While caution is required when extrapolating these findings to humans, particularly sensitive populations such as infants, elderly individuals, and those with respiratory conditions, the present results support the safe use of SAEW in animal models and suggest potential benefits in human environments when applied appropriately.”

Reviewer #2: Major Concerns:

1. The study lacks a detailed discussion on the potential risks of chronic exposure in humans, particularly in sensitive populations.

Answer Reviewer #2-1, Thank you for your comment. We have revised the Discussion to address the potential risks to sensitive populations. While our long-term inhalation study in rats demonstrated no adverse effects associated with SAEW exposure, we acknowledge that caution is required when extrapolating these findings to humans. Sensitive populations—such as infants, elderly individuals, and people with pre-existing respiratory conditions—may respond differently. Nonetheless, our results indicate that SAEW can be used safely in animal models, supporting its potential as an effective disinfection method that could also benefit sensitive populations when applied appropriately (Discussion, Lines 430-437).

2. Although SAEW was found safe in rats, extrapolation to human environments (e.g., schools, hospitals) warrants further discussion.

Answer Reviewer #2-2, Thank you for your comment. We have revised the Discussion to note that while SAEW was found safe in rats, extrapolation to human environments—such as schools or hospitals—warrants caution. Our Discussion emphasizes that sensitive populations may respond differently and that further investigation is necessary. These revisions have been incorporated into the final paragraph of the Discussion (Lines 430-437).

3. The experimental design should mention whether blinding was used during histopathological assessments to minimize observer bias.

Answer Reviewer #2-3, Thank you for your comment. We have clarified the experimental design in the Methods section. A total of 24 rats (12 males and 12 females) were allocated into two groups: an untreated control group and an iPOSH spatial spraying group (n = 6 males and 6 females per group), with animals randomly assigned to each group. All assessments, including blood tests, blood biochemical analyses, bronchoalveolar lavage fluid (BALF) analysis, urine tests, and histological evaluations, were performed by an external service provider who was blinded to the experimental groups and unaware of the treatment conditions, thereby minimizing observer bias (Methods, Line 211-216).

Minor Issues:

1. The graphical representation of data (e.g., colony counts, particle size distribution) could be improved for clarity.

Answer Reviewer #2-Minor-1, Thank you for your helpful comment. We have revised the figures to improve clarity. Specifically, we increased the marker size in Fig. 1B, adjusted the line thickness and style in Fig. 1C, and corrected the Y-axis label in Fig. 2 to “Colony counts (CFU).”

2. Some parts of the text are overly descriptive and could be streamlined to maintain scientific focus.

Answer Reviewer #2-Minor-2, Thank you for your insightful comments. To address your suggestions, we revised and streamlined several overly descriptive passages in the Introduction and removed redundant background information from the Discussion. Specifically, we shortened the general description of viral transmission routes, deleted detailed explanations of SAEW’s antimicrobial mechanism that were not essential for framing our study, and removed repeated descriptions comparing the disinfectant susceptibility of SARS-CoV-2 and E. coli. In the Discussion, we also trimmed sections that reiterated introductory content and focused more directly on the interpretation and implications of our results. These revisions improve readability and maintain a clearer scientific focus.

3. The potential for formation of disinfection byproducts under various environmental conditions should be acknowledged.

Answer Reviewer #2-Minor-3, Thank you for your comment. We have clarified in the Introduction that although disinfection byproducts may form when HOCl comes into contact with organic matter, SAEW is based on water and sodium chloride, and its byproducts are expected to pose minimal toxicity under the experimental conditions used in this study (Lines 64-67).

Reviewer #3: This manuscript investigates the chronic safety of slightly acidic electrolyzed water (SAEW) applied as an aerosol (“spatial spraying”) in Sprague–Dawley rats over a 3-month period. The study evaluates hematological, biochemical, and histopathological parameters to determine whether repeated SAEW exposure induces any toxicological or inflammatory changes.

The topic is relevant to environmental disinfection and public health, particularly in the context of widespread use of SAEW as a disinfectant. The study is generally well organized, and the results are clearly presented. However, several critical methodological details are missing, and the experimental design would benefit from addi

---

## [Decision Letter · Decision Letter 1]

2 Jan 2026

Effective and safe: long-term aerosol disinfection of slightly acidic electrolyzed water causes no harm in rats

PONE-D-25-54069R1

Dear Dr. Seki,

We’re pleased to inform you that your manuscript has been judged scientifically suitable for publication and will be formally accepted for publication once it meets all outstanding technical requirements.

Kind regards,

Jiafu Li, Ph.D

Academic Editor

PLOS One

Additional Editor Comments (optional):

Reviewers' comments:

Reviewer's Responses to Questions

**Comments to the Author**

Reviewer #1: All comments have been addressed

Reviewer #3: All comments have been addressed

Reviewer #4: All comments have been addressed

2. Is the manuscript technically sound, and do the data support the conclusions?

Reviewer #1: Yes

Reviewer #3: Yes

Reviewer #4: Yes

3. Has the statistical analysis been performed appropriately and rigorously?

Reviewer #1: Yes

Reviewer #3: Yes

Reviewer #4: I Don't Know

4. Have the authors made all data underlying the findings in their manuscript fully available?

Reviewer #1: Yes

Reviewer #3: Yes

Reviewer #4: Yes

5. Is the manuscript presented in an intelligible fashion and written in standard English?

Reviewer #1: Yes

Reviewer #3: Yes

Reviewer #4: Yes

Reviewer #1: The authors reasonably replied to all my previous criticisms and comments. The paper was significantly improved. I have no further comments.

Reviewer #3: (No Response)

Reviewer #4: The authors have satisfactorily addressed all reviewer comments. The revisions have improved the clarity of the methodology, strengthened the presentation of results, and enhanced the overall quality of the manuscript. I find the revised version suitable for publication in its current form and recommend acceptance.

**Do you want your identity to be public for this peer review?** For information about this choice, including consent withdrawal, please see our Privacy Policy

Reviewer #1: No

Reviewer #3: No

Reviewer #4: **Yes:**  Magda Elsayed Abdelgawad

---

## [Editor Report · Acceptance letter]

PONE-D-25-54069R1

PLOS One

Dear Dr. Seki,

I'm pleased to inform you that your manuscript has been deemed suitable for publication in PLOS One. Congratulations! Your manuscript is now being handed over to our production team.

Kind regards,

on behalf of

Dr. Jiafu Li

Academic Editor

PLOS One